# Design and Synthesis of a New Soluble Natural β-Carboline Derivative for Preclinical Study by Intravenous Injection

**DOI:** 10.3390/ijms20061491

**Published:** 2019-03-25

**Authors:** Sébastien Marx, Laurie Bodart, Nikolay Tumanov, Johan Wouters

**Affiliations:** 1Department of Chemistry, NAmur MEdicine and Drug Innovation Center (NAMEDIC-NARILIS), University of Namur, 61 rue de Bruxelles, B-5000 Namur, Belgium; laurie.bodart@unamur.be (L.B.); nikolay.tumanov@unamur.be (N.T.); johan.wouters@unamur.be (J.W.); 2URBC—NARILIS, University of Namur, 61 rue de Bruxelles, B-5000 Namur, Belgium

**Keywords:** antiproliferative activity, harmine derivative, IV injection, mechanosynthesis

## Abstract

Harmine is a natural β-carboline compound showing several biological activities, including antiproliferative properties, but this soluble natural molecule lacks selectivity. Harmine derivatives were reported to overcome this problem, but they are usually poorly soluble. Here, we designed and synthesized a new 2, 7, 9-trisubstituted molecule (1-methyl-7-(3-methylbutoxy)-9-propyl-2-[(pyridin-2-yl)methyl]-9*H*-pyrido[3,4-b]indol-2-ium bromide) with a solubility of 1.87 ± 0.07 mg/mL in a simulated injection vehicle. This compound is stable for at least 72 h in acidic and physiological conditions (pH 1.1 and 7.4) as well as in a simulated injection vehicle (physiological liquid + 0.1% Tween80®). Solubility in those media is 1.06 ± 0.08 mg/mL and 1.62 ± 0.13 mg/mL at pH 7.4 and 1. The synthesized molecule displays a significant activity on five different cancer cell lines (IC_50_ range from 0.2 to 2 µM on A549, MDA-MB-231, PANC-1, T98G and Hs683 cell lines). This compound is also more active on cancer cells (MDA-MB-231) than on normal cells (MCF-10a) at IC_50_ concentrations. Due to its high activity at low concentration, such solubility values should be sufficient for further in vivo antitumoral activity evaluation via intravenous injection.

## 1. Introduction

Harmine is a natural β-carboline alkaloid compound (Figure 1) extracted from *Peganum Harmala* seeds and displaying several biological activities. Among them, antidepressant, antifungal, antimicrobial and antiplasmodal properties were reported, as well as an antioxidative effect because of its action on MonoAmine Oxidase (MAO) [1,2,3,4]. Antitumoral activities of harmine were largely studied because it is reported to interact with DNA and to inhibit topoisomerase I, leading to cell death [5,6]. This natural molecule was also reported as an apoptotic pathway activator in hepatocarcinoma because it decreases Bcl-2 protein level without any modification of Bax pro-apoptotic expression [7,8,9]. Harmine also leads to in vitro apoptosis in stomach cancer cells by decreasing the cyclooxygenase-2 (COX-2) expression while increasing Bax protein expression [10] or by inhibiting the Akt phosphorylation, which is essential for cell survival [11].

Due to its activity on several biological targets (MAO, topoisomerase I, COX-2), harmine is considered as a potential drug candidate. However, harmine on itself exhibits poor selectivity, which may cause undesirable effects. Using derivatives of this natural compound might help overcome this issue. Therefore, several harmine derivatives were developed and studied in order to promote antiproliferative properties with the objective to act on the intrinsic apoptosis resistance of tumor cells, one of the hallmarks of cancer [12,13]. Despite a high selectivity for cancer cells and an IC_50_ close to micromolar concentrations, these derivatives are generally difficult to develop as lead compounds for pre-clinical studies because of their poor solubility leading to the impossibility of intravenous injection, which is the reference administration route for in vivo studies [9,12,14,15,16,17,18,19,20]. Pharmacokinetic properties such as solubility or metabolic stability are crucial for drug development as illustrated by the fact that the lack of proper pharmacokinetic properties represents 39% of failure during drug development [21]. In order to improve harmine derivatives solubility while maintaining their antiproliferative activities, cyclodextrins complexation has been already performed in our group and this formulation led to a two-fold increase in solubility at physiological pH [22]. Despite those studies, to the best of our knowledge, only parenteral injection is referred for harmine derivatives administration.

In this work, we designed an original tri-substituted harmine derivative. On the basis of previously published QSAR (quantitative structure-activity relationship) studies [22] and on preliminary kinetic solubility data performed in our group, this new molecule should present improved solubility in comparison to other reported compounds while retaining good biological activity. The designed compound was then synthesized by an original route using mechanochemistry for the last step (trisubstitution). This new compound was then characterized in terms of stability and solubility as it was designed to allow future intravenous injection which was never reported with harmine analogues. The activity of our new derivative was also evaluated on five different cancer cell lines.

## 2. Results

### 2.1. Design

Several QSAR studies were performed in order to improve anti-proliferative activity while decreasing neurotoxic effects [23,24]. In general, 7-9-disubstituted derivatives with branched, non-branched alkyl groups and benzyl groups are less toxic (LD_50_
≥ 100 mg/kg in mice [17]) than harmine (59 mg/kg in mice [18]). However, N^2^–benzyl alkylation of di-substituted harmine derivatives (so reaching tri-substituted compounds) results in acute toxicity higher than harmine (LD_50_
≥ 3.75 mg/kg in mice [17]) but also in higher antitumor activity than harmine and di-substituted analogues. Besides, di- and tri-substituted harmine derivatives enable a loss of neurotoxic effects, which are however observed with harmine [8,18,20]. In 2004, Cao et al. have shown that a short alkyl or benzyl substituent on position 9 of β-carboline core resulted in better antitumor activities [8,18]. The same group demonstrated, in 2013, that methoxy group on position 7 is unfavorable because it results in neurotoxic effects and that the replacement of this group by a bulky alkoxy group eliminates those undesirable effects [16]. Zhang et al have shown that a branched isopentyl group on position 7 enables to decrease toxicity and N^2^-alkylation with benzyl group promotes antitumor activity at low dosage [17]. 

Other 3D-QSAR studies performed in our group, combined with solubility at physiological pH and activity analysis were also performed [24]. This has led to the conclusion that a small alkyl group on N^9^ is adequate because it reduces lipophilicity of the molecule and that the positive charge on N^2^ is highly favorable for the compound activity (Figure 2A). However, a polar substituent on O^7^ and N^9^ results in a loss of the activity despite a better solubility [24]. 

From those data, we chose an isopentyl substituent on position 7, which is suitable to decrease neurotoxicity of the targeted compound. We selected a propyl group on position 9 as a short alkyl substituent, seems to be suitable to maintain good antiproliferative properties. For the third substituent, we chose a methylpyridyl moiety because our preliminary kinetic solubility data performed on mono- and di-substituted intermediates (**1a** and **1b**, Figure 2B) reveals good solubility at low pH associated to protonation of N^2^. Cao et al. showed that a benzyl substituent on N^2^ results in micromolar activity against various cancer cell lines despite low LD_50_ [16]. This high activity can result in low administered doses, thus counter-balancing the low LD_50_ associated with this N^2^ substitution. These results suggest pyridine as a suitable third substituent as it should allow protonation in acidic conditions. It has been shown that pyridine scaffold in lead compounds can result in higher solubility because of the presence of the heteroatom [25,26]. Finally, the pyridyl group offers good potential for salification and/or cocrystallization which can also improve pharmacological properties [27,28,29,30]. Proposed novel trisubstituted harmine with selected substituent is illustrated at Figure 2C.

### 2.2. Synthesis of Compound 2

A synthetic pathway was already described in the literature for the synthesis of mono-, di-, and tri-substituted harmine derivatives [13,24]. Compounds **1a** and **1b** were synthesized according to these procedures. Compounds **1a** was synthesized starting from harmol, obtained by harmine demethylation in acidic condition and at 140 °C (reflux). Monoalkylation of harmol was performed in presence of caesium carbonate and 1-bromo-3-methyl butane at room temperature in *N*,*N*-dimethylformamide (DMF), yielding 67% of pure **1a**. Compound **1b** was synthesized, with a yield of 74%, in DMF, by adding sodium hydride and iodopropane to **1a**. However, synthesis of compound **2** was not achievable by those methods and an original approach was developed (Figure 3A). Compound **1b** was ground with 2-bromomethylpyridine hydrobromide in presence of Na_2_CO_3_ as well as few drops of solvent (liquid-assisted grinding, LAG). Grinding conditions were optimized: absence of solvent vs. grinding in the presence of ethanol (EtOH) and methanol (MeOH). The crude product was then purified by Flash Chromatography with a yield of 48%. Such mechanosynthesis pathway was unprecedented for tri-substituted harmine derivatives synthesis. The solvent used during grinding played a major role as indicated by the yield obtained in the presence of ethanol vs. methanol (21% *vs.* 48%). Details of synthesis optimization are available in Appendix A. Final product and its intermediates were characterized by ^1^H and ^13^C NMR, elemental analysis and single-crystal X-ray diffraction (Figure 3B and Appendix A).

### 2.3. Stability and Solubility of Compound 2 in Three Different Media

Compound **2** solubility and stability measurements were performed at 37 °C at pH 1.1 (0.02 M NaCl, pH 1.1 obtained by HCl addition) to mimic gastric conditions, at pH 7.4 (0.050 M Na_2_HPO_4_/NaH_2_PO_4_ and 0.02 M NaCl, pH 7.4) and simulated injection vehicle (0.9% NaCl and 0.1% Polysorbate 80). Stability measurements were performed from 0 h to 72 h after complete solubilization. UV-spectra reveals no modification at least for 72 h at 37 °C (Figure 4 and Appendix A). These results indicate that compound **2** is stable for at least this duration and so thermodynamic solubility measurement for 72 h can be performed. At pH 1.1 as well as in the simulated injection vehicle, a higher solubility is observed in comparison with the solubility at physiological pH. Solubility is 1.06 ± 0.08 mg/mL at physiological pH, 1.62 ± 0.13 mg/mL at acidic pH, as well as 1.87 ± 0.07 mg/mL in a simulated vehicle injection (Table 1). 

### 2.4. Antiproliferative Activity Determination of Compound 2 on Five Different Cancer Cell Lines

2,7,9-Tri-substituted harmine derivatives were described to have micromolar to submicromolar antiproliferative activity [13,16,24]. Besides this cytotoxic activity on cancer cells, a cytostatic behavior was also observed at the IC_50_ value for some molecules [13,24]. Antiproliferative activity of compound **2** was evaluated using a colorimetric MTT assay for 72 h [24]. To assess eventual cytostatic behavior of compound **2**, cancer cells metabolic activity before incubation with the compound of interest was determined (T = 0 h) (Figure 5). Compound **2** activity was studied on different cancer cell lines: two glioma cell lines (T98G, Hs683), one breast cancer cell line (MDA-MB-231), one lung cancer cell line (A549) and one pancreatic cell line (PANC-1). A mean submicromolar antiproliferative activity was observed (Table 2). A cytostatic behavior at IC_50_ range of concentration for A549 and MDA-MB-231 cell lines was observed, meaning a non-proliferation of cancer cells in presence of compound **2** at IC_50_ concentration, (Figure 5A,B: at IC_50_ concentration there is no increase of metabolic activity in comparison to T = 0 h). Cytotoxicity of compound **2** against normal cells (epithelial breast cells MCF-10a) vs. cancer cells (MDA-MB-231) was also determined. We found that compound **2** is more active on a cancer cell line than on normal cells in the IC_50_ concentration range (Figure 5F).

## 3. Discussion and Conclusions

Some harmine derivatives display a great in vitro activity on cancer cells and present a real interest for the development of new anti-tumor strategies. However, during the clinical trials, these compounds failed to reach the last step due to their limited pharmaceutical properties especially a weak solubility in physiological solutions. In this work, we designed compound **2** on the basis of QSAR and kinetic solubility data of mono- and di-substituted molecules. Compound **2** has been synthesized by an original route implying mechanochemistry for the last synthesis step. This molecule displayed a solubility of 1.87 ± 0.07 mg/mL in a simulated injection vehicle, as well as a significant anticancer activity on five different cell lines (from the submicromolar for A549, MDA-MB-231, PANC-1 and Hs683 cell lines to the micromolar for T98G) by blocking the proliferation of cancer cells. 

We have evaluated the stability of compound **2** in three different conditions (at pH 1.1, at pH 7.4 and in a simulated injection vehicle) for three days and at 37 °C because it is compatible with physiological temperature and because a higher temperature can result in a better solubilization. We observe that the compound remains stable for at least three days in the three different solutions tested, which allowed determination of its thermodynamic solubility. 

Polysorbate 80 plays a major role in the solubilization of compound **2** as 0.1% of this surfactant is sufficient to increase the solubility by a factor of 1.8 in comparison to physiological pH solution. Indeed, up to now, reported in vivo evaluations of harmine derivatives were done by intraperitoneal administration. The solubility of compound **2** is similar to those of antitumoral agents currently available for cancer treatment such as Vincristin® available under perfusion flask with a concentration of 2 mg/mL or Idarubicin® at a concentration of 10 mg/mL [31]. Injection vehicle composed by a minimum of additives has been evaluated here for the solubilization of compound **2**. Other injection vehicles with a higher concentration of polysorbate 80 (up 10% in water) or with other components (surfactant, cyclodextrin, organic co-solvent) could be considered in order to further improve the solubility of the compound of interest [32,33]. 

Solubility results at pH 1.1 suggest protonation of compound **2** at acidic pH. This could eventually lead to a better absorption via per os administration. However, the maximum tolerated a dose of compound **2** and its potential anti-tumoral activity have to be verified before any investigation of compound preparation for eventual future per os administration.

Compound **2** was further evaluated for its antiproliferative properties. Despite a lower activity on T98G cancer cell line, compound **2** demonstrated its best anti-proliferative activity on another glioma cell line (Hs683), with an IC_50_ value under 0.5µM. This molecule displayed a cytostatic effect with a treatment at IC_50_ concentration on two cancer cell lines (A549 lung cancer and MDA-MB-231). Considering activity of compound **2** at low concentrations (IC_50_ ranging from 0.261 to 2.190 µM), we hypothesize that its solubility in simulated vehicle injection (1.87 ± 0.073 mg/mL) should be sufficient to observe in vivo antitumoral activity via intravenous injection, which is the reference route for the chemotherapy treatment. 

Ordinarily, harmine or its derivatives are known to induce apoptosis in cancer cells [3,34,35]. Nevertheless, this trisubstituted harmine derivative could have an alternative mechanism of action as a similar trisubstituted harmine derivative (CM16) has been described with an in vitro antiproliferative activity and a cytostatic anticancer effect by inhibiting the protein synthesis [24]. CM16 rapidly accumulates into cells close to the perinuclear region. Apparently, CM16 does not induce cell cycle arrest nor interfere with DNA as opposed to harmine [36,37]. Thanks to the high structural similarity between CM16 and compound **2,** both compounds could act on cancer cells by the same mechanism of action.

Here, we designed a new derivative of the natural harmine molecule with high activity on different cancer cell lines (submicromolar to micromolar IC_50_ on five cancer cell lines) with enough solubility (1.87 ± 0.073 mg/mL in simulated injection vehicle) to allow IV administration for its further in vivo characterization. 

## 4. Materials and Methods

### 4.1. Cell Culture

MDA-MB-231 human breast cancer cells, T98G human brain cancer cells and PANC-1 human epithelioid carcinoma were maintained in culture in 75-cm^2^ polystyrene flasks (Corning®, Strassen, Luxembourg) with respectively 15 mL of Dulbecco’s modified Eagle’s medium (#341966-029 DMEM 1X; ThermoFisher, Gibco®, Geel, Belgium) containing 4.5g/L of glucose, 10% of fetal calf serum (#10270 ThermoFisher, Gibco®, Geel, Belgium) and supplemented with penicillin-streptomycin 10,000 U/mL (#15140 ThermoFisher, Gibco®, Geel, Belgium) incubated under an atmosphere containing 5% CO_2._ HS683 human glioma and A549 human lung carcinoma were maintained in culture in 75-cm^2^ polystyrene flasks (Corning®) with respectively 15 mL of Roswell Park Memorial Institute medium (Gibco® RPMI 1640, Thermofisher, Geel, Belgium) containing 10% of fetal calf serum (#10270 ThermoFisher, Gibco®, Geel, Belgium) and supplemented with penicillin-streptomycin 10,000 U/mL (#15140 ThermoFisher, Gibco®, Geel, Belgium), glutamine (200 mM) and HEPES (25 mM) incubated under an atmosphere containing 5% CO_2_. MCF-10-a human non-tumorigenic breast epithelial cells were maintained in 75-cm^2^ polystyrene flasks (Corning®, Strassen, Luxembourg) with 15 mL of Gibco®, Geel, Belgium) F-12 Nutrient Mixture (Ham) containing 5% of horse serum (Invitrogen®) supplemented with 10 µg/mL of human Insulin (Sigma®, Schnelldorf, Germany), 0.5 µ/mL of Hydrocortisone (Sigma®, Schnelldorf, Germany) and 20 ng/mL of Epidermal Growth Factor (Peprotech®, London, UK) incubated under an atmosphere containing 5% CO_2_. The cell lines were obtained from the American Type Culture Collection (ATCC, Manassas, VA).

### 4.2. Synthesis

#### 4.2.1. Solvents and Reagents

Solvents from Biosolve (Dieuze France) were used without further purification. All reagents were purchased from Sigma-Aldrich (Schnelldorf, Germany), TCI (Zwijndrecht, Belgium) and Interchim (Mannheim, Germany). All new compounds were determined to be >95% pure by LC-MS (10 µL injection volume at a concentration of 20 µg/mL).

#### 4.2.2. Nuclear Magnetic Resonance

All NMR (^1^H and ^13^C) were recorded on a Jeol spectrometer (JNM EX-400) at 25 °C. Chemical shifts are reported in parts per million (ppm) using the solvent residual peak as a reference (DMSO-d_6_: d_H_: 2.50 ppm, d_C_: 39.52 ppm; CD_3_OD: d_H_: 3.31 ppm, d_C_: 49.00 ppm). Coupling constants (J) are reported in Hertz (Hz). The resonance multiplicity is described as s for singlet, d for doublet, t for triplet, q for quadruplet and m for multiplet. 

#### 4.2.3. Liquid Chromatography Coupled with Mass Spectrometry

LC-MS analyses were performed on an Agilent 1100 series HPLC coupled with an MSD Trap SL system, using detection at 254 nm. The apparatus was equipped with an Agilent Zorbax Sb-C18 (3.0 mm × 100 mm; 3.5 µm) separation column in acetic acid 0.1% and acetonitrile as eluent with a flow rate of 0.5 mL/min (Method B). Masses were determined using electron spray ionization (ESI) in positive mode. 

#### 4.2.4. Elemental Analysis

Elemental analysis was performed on a FlashEA 1112 series organic elemental analyzer (C, H, N). 

#### 4.2.5. Thin-Layer Chromatography and Flash Column Chromatography

Thin-layer chromatography was performed on silica gel plates (silica gel GF254, VWR) and revelation was made using UV lamp with a wavelength of 254 nm. Column chromatography was performed in a BiotageSP1 25 M column equipped with a UV spectrophotometer as a detector (wavelengths of 254 and 320 nm).

#### 4.2.6. Mechanochemistry

The mechanochemical synthesis was conducted in 2 mL Eppendorf tubes with seven stainless steel balls (1 mm in diameter) using a Retsch MM 400 shaker mill with methanol as solvent. 

#### 4.2.7. Melting Point Measurement

Melting points were measured on a Buchi Melting Point B540 apparatus in open capillaries.

#### 4.2.8. Single-Crystal X-ray Diffraction (SCXRD)

Selected crystals of suitable size were mounted on an Oxford Diffraction Gemini Ultra R system (4-circle kappa platform, Ruby CCD detector) using Mo $K_{\alpha }$ (λ = 0.71073 Å) for compounds **1b** and **2** or Cu $K_{\alpha }$ (λ = 1.54184 Å) for compound **1a**. Structures were solved using SHELXT [38] and then refined using SHELXL-2018/1 [39] within Olex [40] and ShelXle [41]. Non-hydrogen atoms were anisotropically refined using a mixture of constrained (for disordered parts of the structure) and independent refinement. Hydrogen atoms implied in strong hydrogen bonds were localized by Fourier difference maps. Structure of **1a** and **1b** are disordered (isopentyl group for **1a** and isopentyl as well as propyl groups for **1b**). Structure **1a** presents accessible voids but no electron density is present inside those voids indicating that the H-bonds are sufficiently strong to maintain the structure in absence of any solvent. Moreover, those voids are isolated (they do not form any channels). Structure **2** is a non-stoichiometrically hydrated salt with water having a refined occupancy of 0.127(9), because of this occupancy and the lack of H-bond donors and acceptors around the water oxygen atom, water hydrogen atoms could not be assigned unambiguously. 

#### 4.2.9. Synthesis of 1-methyl-9*H*-pyrido[3,4-b]indol-7-ol (harmol)

Harmine (5.0 g, 20.1 mmol) was dissolved in a 1:1 volume mixture of acetic acid (+99%) (65 mL) and hydrobromic acid (> 48%) (65 mL). The mixture was stirred and heated up to 140 °C for 2 days. The reaction was followed by thin layer chromatography (TLC) (85:15 dichloromethane/ethanol). At the end of the reaction, the reaction mixture was extracted using distilled water and the precipitate was filtered. The residual filtrate was evaporated under vacuum at 60 °C. Yield = 98%. Rf: 0.11 (dichloromethane/ethanol 85/15). ^−1^H NMR (DMSO-d_6_) δ: 2.92 (s; 3H; **CH_3_**); 6.88 (dd; J_6-8_ = 1.37 Hz; J_5-6_ = 7.33 Hz; 1H, **H-6**); 7.00 (d; J_6-8_ = 1.60 Hz; 1H; **H-8**); 8.23 (d; J_3-4_ = 8.70 Hz; 1H; **H-4**); 8.34 (q, J_5-6_ = 6.18 Hz; J_3-4_ = 9.50 Hz 2H **H-5**; **H-3**); 10.4 (s, 1H; O**H**), 12.5 (s, –N**H**, 1H). Elemental analysis: calculated for C_12_H_13_N_2_O_2_: C; 48.50%; H; 4.41%; N; 9,43%; Br, 26.89; found: C; 48.21%; H; 4.36%; N; 10.65%.

#### 4.2.10. Synthesis of 1-methyl-7-(3-methylbutoxy)-9*H*-pyrido[3,4-b]indole (**1a**)

Compound **1a** was synthesized according to the general procedure described in Reference [24] from 1-methyl-7-hydroxy- β-carboline (harmol) (1.97g; 7.04 mmol) in the presence of caesium carbonate (7.20 g; 22.2 mmol) and 1-bromo-3-methylbutane (1.33 mL; 1.1 mmol) in 40 mL of DMF. White solid was obtained with a yield of 67%, Rf: 0.36 (dichloromethane/ethanol 85/15). For single-crystal X-ray diffraction, the powder was recrystallized in dichloromethane. ^1^H NMR (DMSO-d_6_) δ: 0.92 (d; J = 6.64 Hz; 6H, CH–**(CH_3_)_2_**); 1.64 (q; J = 6.41 Hz; 2H; **CH_2_**–CH–(CH_3_)_2_); 1.75–1.85 (m; 1H; **CH**–(CH_3_)_2_; 2.68 (s; 3H; **CH_3_**); 4.06 (t; J = 6.64 Hz; 2H; CH_2_–O); 6.79 (dd; J_6-8_ = 1.37 Hz; J_5-6_ = 8.70 Hz; 1H, H-6); 6.96 (d; J_6-8_ = 2.06 Hz; 1H; H-8); 7.75 (d; J_3-4_ =5.27 Hz; 1H; H-4); 7.99 (d; J_5-6_ = 8.47 Hz; 1H; H-5); 8.10 (d; J_3-4_ = 5.27 Hz; 1H; H-3). ^13^C NMR (DMSO-d_6_) δ: 20.86; 23.02; 25.18; 37.98; 66.64; 95.75; 109.94; 112.43; 115.29; 123.11; 127.74; 135.05; 138.25; 141.25; 142.45; 159.96. MS: [M + H]^+^ = 269.5. T_melting_ = 230, 7–232, 7 °C. Elemental analysis: calculated for C_17_H_20_N_2_O: C; 76.09%; H; 7.51%; N; 10.44%; found: C; 76.66%; H; 7.51%; N; 10.05%.

#### 4.2.11. Synthesis of 1-methyl-7-(3-methylbutoxy)-9-propyl-9*H*-pyrido[3,4-b]indole (**1b**)

Compound **1b** was synthesized according to the general procedure described in reference [24] from **1a** (0.601 g; 2.24 mmol) in the presence of sodium hydride (0.25 g, 11.2 mmol) and iodopropane (0.33 mL; 3.35 mmol) in 42 mL of DMF. After purification and evaporation of organic, a crystalline beige solid was obtained and single crystals of adequate size were present and used for SCXRD. The crystalline beige powder was obtained, yield = 74%. Rf: 0.68 (dichloromethane/ethanol 85/15). ^1^H NMR (DMSO-d_6_) δ: 0.89(t; J = 7.33 Hz; 3H CH_2_**–CH_3_**); 0.93 (d; J = 6.64 Hz;6H; (CH–**CH_3_**)**_2_**); 1.65 (q; J = 6.64 Hz; 2H; **CH_2_**–CH_3_); 1.72(q; J = 7.56 Hz; 2H **CH_2_**–CH–(CH_3_)_2_); 1.81 (m; 1H; **CH**–(CH_3_)_2_); 2.89 (s; 3H; CH_3_); 4.11 (t; J = 6.64 Hz; 2H; CH_2_–O); 4.47 (t; J = 7.56 Hz; 2H; CH_2_–N); 6.82 (dd; J_6-8_ = 2.06 Hz; J_5-6_ = 7.67 Hz; 1H; H-6); 7.16 (d; J_6-8_ = 1.83 Hz 1H; H-8); 7.82 (d; J_3-4_ = 5.27 Hz; 1H; H-4); 8.02 (d; J_5-6_ = 8.47 Hz; 1H; H-5); 8.12 (d; J_3-4_ = 5.27 Hz; 1H; H-3). ^13^C NMR (DMSO-d_6_) δ: 11.48; 23.05; 23.58; 24.14; 25.15; 45.87; 66.85; 94.88; 109.91; 112.73; 114.60; 122.85; 128.91; 135.13; 138.22; 140.99; 143.37; 160.42. MS: [M + H]^+^ = 311.4. T_melting_ = 82.2–83.3 °C. Elemental: analysis, calculated for C_20_H_26_N_2_O: C; 77.38%; H; 8.44%; N; 9.02%; found: C; 77.15%; H; 8.57%; N; 8.65%.

#### 4.2.12. Synthesis of 1-methyl-7-(3-methylbutoxy)-9-propyl-2-[(pyridin-2-yl)methyl]-9*H*-pyrido[3,4-b]indol-2-ium bromide (**2**)

Compound **2** was synthesized by mechanochemistry. A solid mixture of compound **1b** (151 mg, 1 equivalent, 0.486 mmol), 2-bromomethylpyridine hydrobromide (123 mg, 1 equivalent, 0.486 mmol) and sodium carbonate (53 mg, 1 equivalent, 0.500 mmol) was ground for 90 min at 30Hz in presence or absence of liquid (30 µL MeOH or 30 µL EtOH) and the mixture was homogenized every 30 minutes. At the end of the grinding, the crude solid mixture was extracted in dichloromethane. The organic layer was washed with distilled water, brine and dried on magnesium sulfate. The organic layer was evaporated under vacuum and the crude product was purified via column chromatography (dichloromethane (DCM)/EtOH, 100:0 to 80:20). White solid was obtained; yield = 48%. Rf = 0.50 (DCM/EtOH 80/20). Crystals suitable for SCXRD measurement were obtained by recrystallization in MeOH and by slow evaporation at ambient temperature. ^1^H NMR (Methanol-d_4_) δ: 1.01 (m; 9H; CH–**(CH_3_)_2_**; CH_2_–**CH_3_**); 1.76 (q; J = 6.64 Hz; 2H; **CH_2_**–CH_3_); 1.90 (m; 3H; **CH_2_**–**CH**–(CH_3_)_2_); 3.14(s; 3H; CH_3_); 4.24(t; J = 6.64Hz; 2H; O–CH_2_); 4.63(t; J = 7.79Hz; 2H; N–CH_2_); 6.10(s; 2H; N–CH_2_ (pyr)); 7.08 (dd;J_5-6_ = 8.93 Hz; J_6-8_ = 2.06 Hz; 1H; H-6); 7.23 (d; J_6-8_ = 1.83Hz; 1H; H-8); 7.36(dd; J_2-3_ = 5.04Hz; J_3-4_ = 6.87Hz; 1H; H-3(pyr)); 7.52 (d; J_5-6_ = 8.01; 1H; H-5); 7.89 (td; J_2-4_ = 1.60Hz J_3-4_ = 7.66Hz; 1H; H-4 (pyr)); 8.25 (d; J = 8.70Hz; 1H; H-5 (pyr)); 8.38 (d; J_3-4_ = 6.41Hz; 1H; H-4); 8.44 (d; J_2-3_ = 4.81Hz; 1H; H-2(pyr)); 8.54 (d; J_3-4_ = 6.64Hz; 1H; H-3). NMR ^13^C (DMSO-d_6_) δ: 9.86; 15.56; 21.62; 23.59; 24.96; 37.72; 60.72; 67.05; 93.70; 112.99; 113.64; 113.74; 122.02; 123.66; 123.99; 134.27; 135.10; 135.71; 139.59; 137.77; 148.34; 149.68; 153.33; 164.09;MS: [M+H]^+^ = 402.3. Elemental analysis: calculated for C_26_H_32_N_3_OBr.H_2_O: C; 62.40%; H; 6.85%; N; 8.40%; Br, 15.97; found: C; 62.17%; H; 6.54%; N; 8.57%.

### 4.3. Determination of Kinetic Solubility of Compounds 1a and 1b at Physiological and Acidic pH

The kinetic solubility of compounds **1a** and **1b** was determined by HPLC method, using a Multiscreen Solubility Filter 96-well plate (Merck®) using a test adapted from [42,43]. First, the calibration curve was performed from five solutions (3.13 µM 12.5 µM, 50 µM, 200 µM and 500 µM, with a final DMSO concentration of 5%). The kinetic solubility was measured in triplicate at physiological pH in phosphate buffer (0.05 M Na_2_HPO_4_; 0.02 M NaCl adjusted at physiological pH using solution NaH_2_PO_4_ 0.05 M; NaCl 0.02 M) and in acidic condition (0.2 M HCl, 0.03 M NaCl adjusted at pH = 1.1 using HCl 37%). In a testing plate, 10 µL of tested compound (10 mM in 100% DMSO) is added to 190 µL of phosphate buffer (5% final concentration in DMSO). After agitation for 90 minutes, the plate is filtered using Millipore collector at 10–12 mmHg for 30 to 60 s. One hundred and sixty microliter of filtrate was transferred in a 96-well plate and 40 µL of acetonitrile is added. After stirring for 5 min at 200–300 rpm at room temperature, the filtrates are directly injected without any filtration in HPLC.

### 4.4. Stability and Solubility Study of Compound **2**

#### 4.4.1. Evaluation of Compound 2 Stability at Physiological pH and in Simulated Injection Vehicle

A solution of compound **2** at a concentration of 1.0 mg/mL was prepared in a buffer at acidic pH (0.02 M NaCl, pH 1.1 obtained by HCl addition), as well as in a buffer at physiological pH (buffer 0.050 M Na_2_HPO_4_/NaH_2_PO_4_ and 0.02 M NaCl, pH 7.4) and a third solution at the same concentration was also prepared in a simulated injection vehicle (0.9% NaCl and 0.1% Polysorbate 80 pH 7.4). Absorbance at 330 nm of the solution was measured after three days of stirring (750 rpm) at 37° (adequate dilution were performed to stay in Beer-Lambert conditions). For the stability study, compound **2** was dissolved in the three different media and its spectra measured at different time (0 h, 2 h, 5 h, 24 h, 48 h and 72 h) between 270 and 450 nm. The stock solution was diluted to stay in Beer-Lambert conditions for the measurement.

#### 4.4.2. Evaluation of the Thermodynamic Solubility of Compound 2

Solubility of compound **2** was evaluated at 37 °C, at pH 1.1 (0.02 M NaCl, pH 1.1 obtained by HCl addition), at physiological pH (buffer 0.050 M Na_2_HPO_4_/NaH_2_PO_4_ and 0.02 M NaCl, pH 7.4) as well as in a simulated injection vehicle (0.9% NaCl and 0.1% Polysorbate 80 pH 7.4). To this aim, an excess of compound **2** was placed in the two different media in a 24 well plate and stirred (1200 rpm) 72 h at 37 °C after hermetic sealing. After 72 h of stirring, the solutions were filtered (0.2 µm) and diluted 100 times. The solubility was determined by UV absorbance at 330 nm using a calibration curve (five points in a range of concentration between 0.005 and 0.1 mg/mL). All measurements were made in triplicate and adequate dissolutions were performed to stay in Beer-Lambert conditions.

### 4.5. Determination of Antiproliferative Activity by 3-(4,5-dimethylthiazol-2-yl)-2,5-diphenyltetrazolium bromide (MTT) Assay

Antiproliferative activity and overall cytotoxicity of compound **2** on cancer and normal cells lines (breast cancer cell line MDA-MB231 and non-tumorigenic breast epithelial cells (MCF-10a) were determined in vitro using colorimetric MTT (Sigma-Aldrich (Schnelldorf, Germany)) solution (0.5 mg/mL in PBS) assay according to a protocol adapted from [44]. Cells were seeded in 96-well culture plate (1200 cells/well 1 day before incubation and were treated for 72 h in culture medium at 37 °C with different concentrations of compound **2** ranging from 1 nM to 10 µM, with semilog concentration increasing of the compound. The yellow MTT product was converted to a purple formazan derivative through mitochondrial enzymatic reduction. At the end of the incubation of 4 h, the plate was centrifugated at 1200 rpm for 5 min and the MTT solution was decanted. Then, blue formazan was solubilized with DMSO and absorbance at 570 nm, which is directly proportional to the number of metabolic active cells, is determined. Experiments were carried out in sextuplicates.

## Figures and Tables

**Figure 1 ijms-20-01491-f001:**
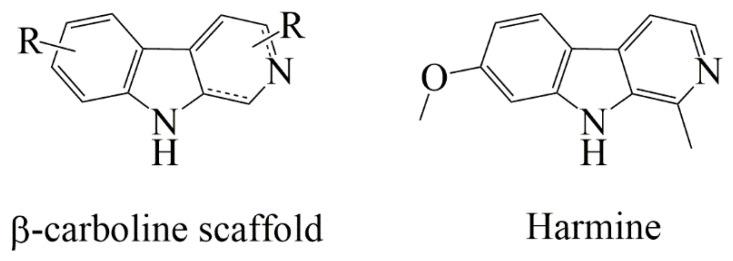
Structure of β-carboline scaffold and of the natural molecule harmine.

**Figure 2 ijms-20-01491-f002:**
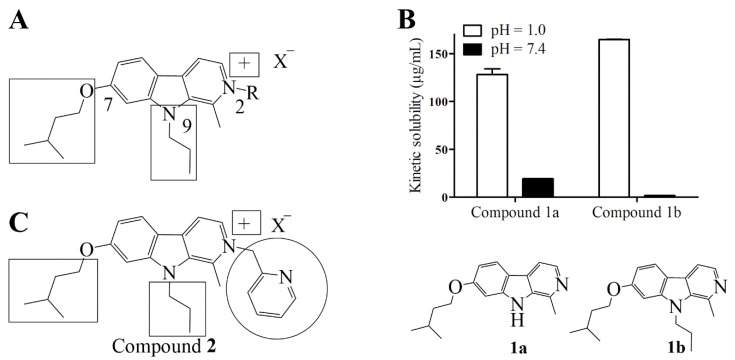
Design of a new β-carboline derivative. (**A**) Important structural elements to maintain antiproliferative activity (highlighted by black frames). (**B**) Preliminary kinetic solubility data of mono- and di-substituted compounds 1a and 1b. (**C**) Proposed novel trisubstituted harmine derivative keeping structural elements necessary to maintain good antiproliferative activity (black frames) while adding a pyridyl moiety that should lead to improve solubility (black circle).

**Figure 3 ijms-20-01491-f003:**
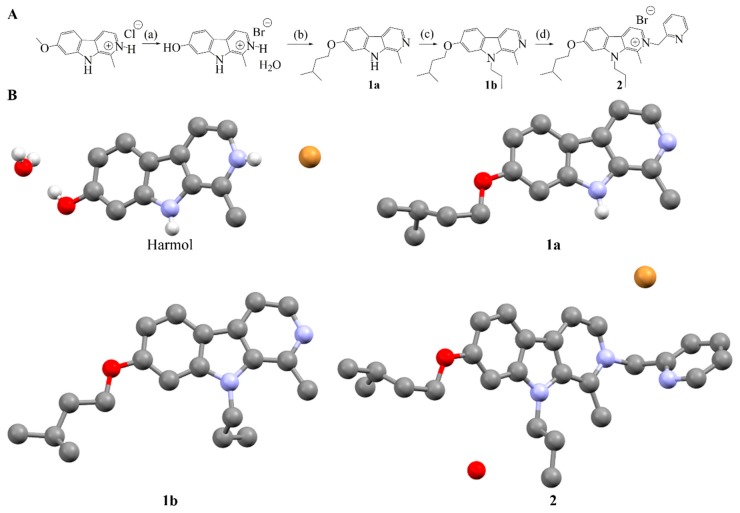
(**A**) (a) Acetic acid (HAc), HBr, 140 °C, 48 h; (b) CsCO_3_, 1-bromo-3-methyl butane, DMF; (c) NaH, iodopropane, DMF; (d) 2-bromomethylpyridine hydrobromide, Na_2_CO_3_ (neat or LAG EtOH or MeOH) and (**B**) Crystal structures of synthesized compounds. Hydrogens of carbon atoms and disorder are not shown for clarity.

**Figure 4 ijms-20-01491-f004:**
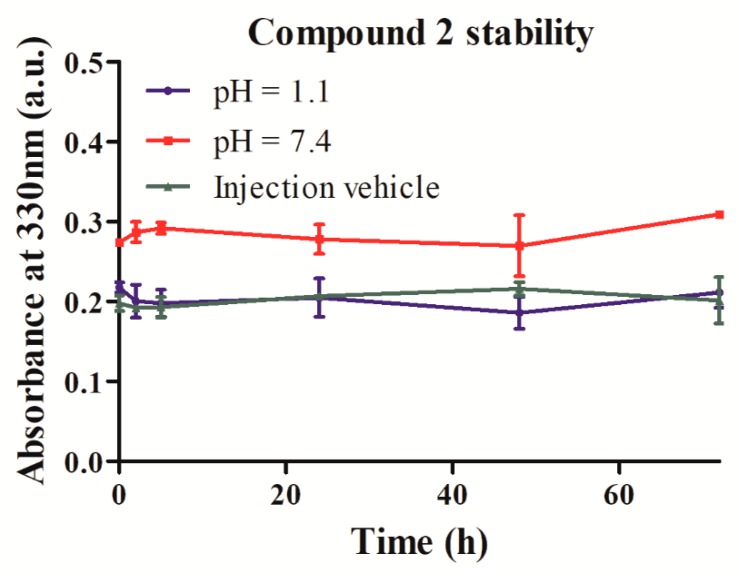
Compound **2** stability for 72 h at 37 °C in pH 1.1, in pH 7.4 and in an injection vehicle composed of 0.9% NaCl and 0.1% Polysorbate 80 pH 7.4.

**Figure 5 ijms-20-01491-f005:**
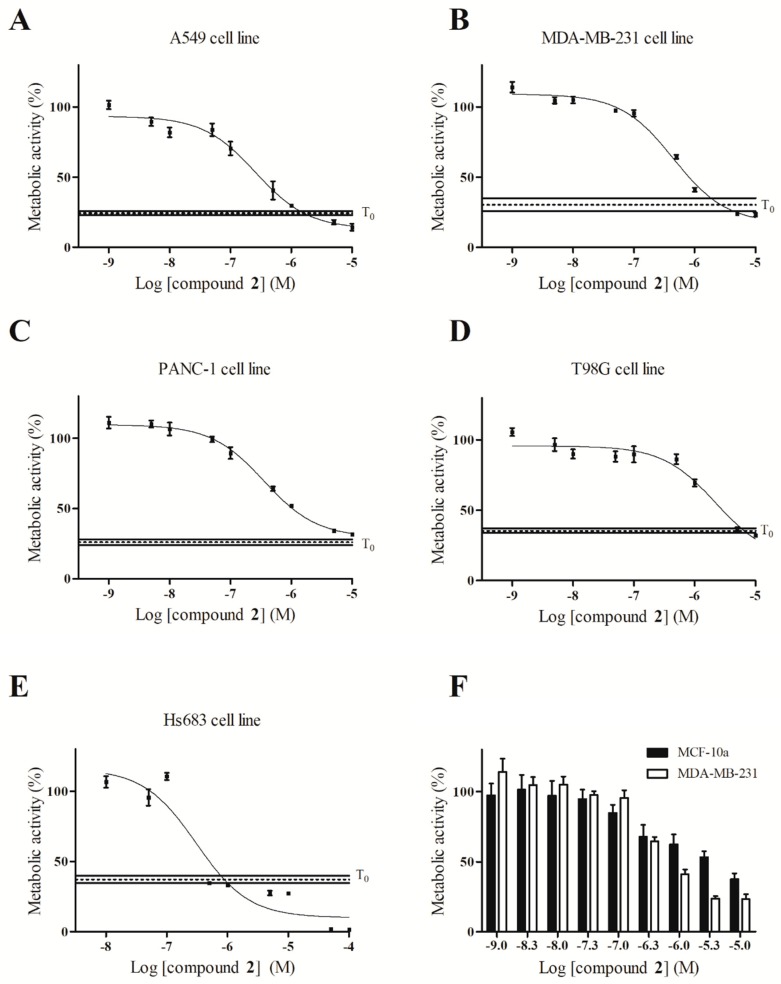
(**A**–**E**): IC_50_ graphs of compound **2** on five different cell lines. Plain horizontal lines (T_0_) represent metabolic activity range just before treatment with compound **2** (metabolic activity at IC_50_ concentration falling in this range means cytostatic behavior of compound **2**). (**F**): Activity of compound **2** on normal (MCF-10a) and breast cancer (MDA-MB-231) cell lines.

**Table 1 ijms-20-01491-t001:** Thermodynamic solubility of compound **2** in those three different media after 72 h at 37 °C.

	pH 1.1	pH 7.4	Injection Vehicle
Thermodynamic solubility at 37 °C (mg/mL) replicate 1	1.46 ± 0.01	0.98 ± 0.01	1.95 ± 0.01
Thermodynamic solubility at 37 °C (mg/mL) replicate 2	1.63 ± 0.02	0.99 ± 0.01	1.90 ± 0.04
Thermodynamic solubility at 37 °C (mg/mL) replicate 3	1.77 ± 0.01	1.15 ± 0.01	1.77 ± 0.01
Mean thermodynamic solubility at 37 °C (mg/mL)	1.62 ± 0.13	1.06 ± 0.08	1.87 ± 0.07

**Table 2 ijms-20-01491-t002:** Determination coefficient, IC_50_ values, confidence interval and cytostatic behavior at IC_50_.

Cell Lines	A549	MDA-MB-231	PANC-1	T98G	Hs683
R^2^	0.97	0.97	0.96	0.88	0.90
IC_50_ (µM)	0.261	0.449	0.350	2.190	0.305
Confidence interval at 95% (µM)	0.205 to 0.331	0.358 to 0.563	0.271 to 0.468	1.24 to 3.89	0.195 to 0.479
Cytostatic at IC_50_	Yes	Yes	No	No	Yes

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
