# Peer review of "Design and Synthesis of a New Soluble Natural β-Carboline Derivative for Preclinical Study by Intravenous Injection"

_ijms, 2019, doi:10.3390/ijms20061491_

Reviewer 1 Report

In this work the authors report the design and the synthesis of a new carboline alkaloid with the aim to improve the lead compound solubility. For this compound several biological tests were also performed showing its activity on different cancer cell lines.

The manuscript is clearly presented and well organized, but minor revisions are required before its publication.

In particular:

The synthesis of compound 2 (lines 107-117) should be better explained according to figure 3A;

In Harmol synthesis paragraph, page 9, lines 278,279, the ratio and mL of the used acids should be inserted;

For compound 1a, page 10, the yield should be inserted.

Author Response

Dear reviewer,

We are grateful for the time you have given to reviewing our work. Please find in the following our responses to your comments.

-          “The synthesis of compound 2 (lines 107-117) should be better explained according to figure 3A.”

o   Synthesis of compound 2 and its intermediates was described with more details (modifications highlighted in yellow):

 A synthetic pathway was already described in the literature for the synthesis of mono-, di-, and tri-substituted harmine derivatives [24,31]. Compounds 1a and 1b were synthesized according to these procedures. Compounds 1a was synthesized starting from harmol, obtained by harmine demethylation in acidic condition and at 140°C (reflux). Monoalkylation of harmol was performed in presence of caesium carbonate and 1-bromo-3-methyl butane at room temperature in dimethylformamide, yielding 67% of pure 1a. Compound 1b was synthesized in DMF, by adding sodium hydride and iodopropane to 1a. However, synthesis of compound 2 was not achievable by those methods and an original approach was developed (Figure 3A). Compound 1b was ground with 2-bromomethylpyridine hydrobromide in presence of Na2CO3 as well as few drops of solvent (liquid-assisted grinding, LAG). Grinding conditions were optimized: absence of solvent vs. grinding in presence of EtOH and MeOH. The crude product was then purified by Flash Chromatography with a yield of 48 %. Such mechanosynthesis pathway was unprecedented for tri-substituted harmine derivatives synthesis. The solvent used during grinding played a major role as indicated by the yield obtained in presence of ethanol vs methanol (21% vs 48%). Details of synthesis optimization are available in Table S1. Final product and its intermediates were characterized by 1H and 13C NMR, elemental analysis and single-crystal X-ray diffraction (Figure 3B and Table S2).

-          “In Harmol synthesis paragraph, page 9, lines 278,279, the ratio and mL of the used acids should be inserted.”

o   Ratios and volumes were inserted (modifications highlighted in yellow):

Harmine (5.0 g, 20.1 mmol) was dissolved in a 1/1 volume mixture of acetic acid (+99%) (65 mL)  and hydrobromic acid (> 48%) (65 mL).

-          “For compound 1a, page 10, the yield should be inserted

o   The yield was inserted (modifications highlighted in yellow):

Compound 1a was synthesized according to the general procedure described in [24] from 1-methyl-7-hydroxy- β-carboline (harmol) (1.97g; 7.04 mmol) in the presence of caesium carbonate (7.20 g; 22.2 mmol) and 1-bromo-3-methylbutane (1.33 mL; 1.1 mmol) in 40 mL of DMF. White solid was obtained with a yield of 67%.

Kind regards,

Sébastien Marx

Reviewer 2 Report

Marx et al. report here the synthesis of a new tri.-substituted soluble harmine derivative that lends itself to future further antitumor activity studies via intravenous injection. This should overcome the poor solubility of previously synthesized harmine derivatives that prevents activity studies by intravenous injection.

Based on evidence reported by other groups, or by the same group, a good rational is furnished for the planned insertion of the new substituents at the 2, 7 and 9 positions of the b-carboline scaffold.  In this respect the literature reported appears pertinent and wide.

The introduction well highlights the problem addressed in the present study.

Solubility and stability studies have been also carried out on the synthesized compound and, interestingly, the prepared compound retains a significant activity on five cancer cell lines.

Overall, I believe that this is a good paper that deserves to be published without modifications.

Author Response

Dear reviewer,

We are grateful for the time you have given to reviewing our work entitled “Design and synthesis of a new soluble natural β-carboline derivative for preclinical study by intravenous injection” by Marx Sébastien, et al..

Kind regards,

Sébastien Marx